# VARIATIONAL BAYES CLASSIFIER

## ABSTRACT

Classifiers have traditionally been designed as fully-observed models. These classifiers are generally deterministic, so they are able to obtain a single output per input. The problem with this is that in this scenario it is not usually possible to capture the model uncertainty. On the other hand, Bayesian models offer the ability to capture this uncertainty, but usually have a higher computational cost. In this paper we propose to build a classifier as a latent variable model. This latent variable corresponds to what is usually called embedding and with our proposal we can model its distribution, which has two fundamental advantages. The first is that by knowing the distribution of the embeddings, the uncertainty of the predictions can be estimated. In addition, certain conditions can be imposed on the distribution of the embeddings to favor aspects such as interclass separation. We also propose an evidence lower bound to optimize the parameters of this classifier which can be maximized using stochastic gradient methods. Finally, we give two alternatives to implement these models using neural networks and demonstrate empirically the theoretical advantages of our proposal using different architectures and datasets.

## 1    INTRODUCTION

Classification is one of the most widespread tasks in the field of Deep Learning. Neural networks have become widely popular in recent years as a common approach for solving this challenge. Krizhevsky et al. (2017); Chen et al. (2021). The problem with these is that, in general, they do not allow capturing predictions uncertainty Guo et al. (2017); Wilson & Izmailov (2020); Gawlikowski et al. (2021). Typically, softmax outputs are misinterpreted as model confidence, but a prediction can be uncertain even though the softmax output is high. Building models that allow measuring the uncertainty of the predictions is of utmost importance, as they provide insight into how reliable these predictions are Gal & Ghahramani (2016). Bayesian Neural NetworksMacKay (1992); Hinton & van Camp (1993); Blundell et al. (2015); Jospin et al. (2022) allow the uncertainty of these predictions to be calculated, but they usually have a high computational cost and their training is problematic.

On the other hand, one of the reasons why classifiers are widely used is because they allow, not only to solve the classification task itself, but also to obtain embeddings that can be useful for other purposes by removing the last layer of the classifier Dohi et al. (2022); Mehta et al. (2022). To solve this task, cross-entropy is typically used as a loss function. However, modifications of this function have been proposed that enhance the performance of classification or enforce specific conditions on the embeddings that are interesting for certain purposes.

Here we present a classifier as a latent variable model, which, rather than extracting one embedding per input, it allows for estimating the distribution of the embeddings. This has numerous advantages, among which the most important are: (i) its derivation permits to impose in a natural way conditions on the distributions of the embeddings, for example, to make them more discriminative between classes, which translates into a better classification performance, (ii) it enables the measurement of uncertainty by allowing the generation of multiple embeddings per input, thereby providing the capability to obtain multiple predictions per input, and (iii) the utilization of the embedding space is optimized in comparison to a traditional classifier.

To model this classifier using neural networks, we propose two alternatives. The first one allows modeling more expressive distributions, but requires slightly modifying the architecture of a con-

ventional classifier. The second, however, models a less expressive distribution, but does not modify the architecture of a conventional classifier at all. The code to implement and train these models is available online [1].

## 2 RELATED WORK

As previously explained, neural networks do not allow, in general, to represent the uncertainty of predictions. As an alternative, extensive research has been conducted on Bayesian Neural Networks (BNN), which enable obtaining the distribution of the weights. This approach offers robustness to overfitting and allows to measure the uncertainty over the predictions of the model. Some popular approaches to these networks include Laplace approximation MacKay (1992); Ritter et al. (2018); Daxberger et al. (2021), stochastic gradient MCMC Welling & Teh (2011); Ma et al. (2015), variational inference Graves (2011); Blundell et al. (2015), expectation propagation Hernández-Lobato & Adams (2015), ensemble of different models Osband et al. (2016); Lakshminarayanan et al. (2017) or Monte Carlo Dropout Gal & Ghahramani (2016). The latter consists of applying dropout in inference and stands out from the rest, since it offers good results, its training is simple and does not add parameters with respect to a classical neural network. However, the prediction time is considerably longer, since it is necessary to pass the inputs to the model several times to obtain well-calibrated predictions. In Riquelme et al. (2018) it is proposed to apply dropout only in the last layer of the model, which speeds up the inference process. On the other hand, other non-Bayesian solutions have been proposed to train calibrated models, such as a temperature scaling of the predictions using a validation dataset Platt et al. (1999); Guo et al. (2017); Minderer et al. (2021).

Other methods try to model the distribution of latent variables. This methodology is called meanfield Hinton & van Camp (1993); Neal & Hinton (1998). These approaches approximate a full posterior distribution by maximizing a lower bound on the marginal likelihood. This requires the ability to integrate a sum of terms in the log joint likelihood using this distribution and often not all these integrals are in closed form. To solve this problem, algorithms based on stochastic approximations have been proposed to estimate such integrals Paisley et al. (2012); Hoffman et al. (2013); Kingma & Welling (2013); Rezende et al. (2014); Titsias & Lázaro-Gredilla (2014). These algorithms have focused on regression problems with the goal of serving as generative models. Our proposal adopts a mean-field methodology, but with the objective of constructing a classifier. To accomplish this objective, we additionally provide a lower bound and a stochastic algorithm for its optimization.

The lower bound we propose provides in a natural way the ability to impose conditions on the latent space (which is the space of the embeddings). These conditions may depend, for example, on the class of the input. A variety of methods have also tried to impose certain conditions on the embeddings. Most of them force inter-class separation and intra-class clustering by modifying the loss function. Among them, contrastive Hadsell et al. (2006) and triplet Schroff et al. (2015) loss have gained considerable popularity. The problem with these is that they require negative data selection processes, which are time-consuming and performance-sensitive. Similarly, methods have been proposed that seek to increase the Euclidean margins between prototypes of different classes Wen et al. (2016); Qi & Su (2017); Zhang et al. (2020). Others focus on increasing the angle between embeddings of different classes Liu et al. (2016; 2018); Deng et al. (2019); Wang et al. (2018). In addition, other ideas suggest to weight gradients according to the difficulty of each training example Lin et al. (2017); Ryou et al. (2019). Finally, there are also works that impose orthogonality between embeddings of different classes on the mini-batch level Ranasinghe et al. (2021)

## 3 LATENT VARIABLE CLASSIFIER

In a classification problem, we assume that the observed variables $\boldsymbol{x}$ (input) and $\boldsymbol{y}$ (class label) follow a distribution $p^*(\boldsymbol{y}|\boldsymbol{x})$ that is unknown and that the objective is to estimate $\boldsymbol{\theta}$ parameters that satisfy that $p_{\boldsymbol{\theta}}(\boldsymbol{y}|\boldsymbol{x}) \approx p^*(\boldsymbol{y}|\boldsymbol{x})$. To perform this estimation, we have a dataset *i.i.d.* $\mathcal{D} = \{\boldsymbol{x}, \boldsymbol{y}\}_{i=1}^{N}$. According to the maximum likelihood (ML) criterion, the unknown parameters of the model are obtained by minimizing $CE\left(q_{\mathcal{D}}(\boldsymbol{x}, \boldsymbol{y}), p_{\boldsymbol{\theta}}(\boldsymbol{y}|\boldsymbol{x})\right)$, where $CE(\cdot)$ is the well-known cross-entropy function Kullback & Leibler (1951) and $q_{\mathcal{D}}(\boldsymbol{x}, \boldsymbol{y})$ is the empirical distribution of the dataset $\mathcal{D}$.

---

[1]Repository url is omitted to maintain anonymity. Code is given in the supplementary material.

## 3.1 Classifier as a Latent Variable Model

Typically, classifiers have been constructed as fully-observed directed models and the cross-entropy between $q_\mathcal{D}(\boldsymbol{x}, \boldsymbol{y})$ and $p_{\boldsymbol{\theta}}(\boldsymbol{y}|\boldsymbol{x})$ has been used as a loss function. However, these models do not allow to calculate or estimate the uncertainty of the predictions, since they return a single output per input and not their distribution or an approximation of them. To solve this problem and allow to measure the uncertainty in a classifier, we propose to model the classifiers as directed models with a latent variable $\boldsymbol{z}$, which we call embedding. Thus, if we know the distribution of the embeddings corresponding to a given input, we can estimate the distribution of predictions and their uncertainty. And all this without the need to resort to Bayesian models, which in many cases have a prohibitive computational cost and are hard to train.

Furthermore, in many cases, classifiers are used as embeddings extractors by removing the last layer of the network. This is because the output of this penultimate layer contains embedded information from the input $\boldsymbol{x}$ that can serve other purposes such as out-of-distribution detection Yang et al. (2021), biometric recognition Bai & Zhang (2021); Minaee et al. (2023), etc. Multiple loss functions alternative to cross-entropy and imposing desired conditions for certain purposes on embeddings have been proposed. The formulation presented in this work also allows imposing conditions on the embeddings distributions and organizing them in the latent space according to the desired task.

With this formulation, we have that $p_{\boldsymbol{\theta}}(\boldsymbol{y}|\boldsymbol{x}) = \mathbb{E}_{p_{\boldsymbol{\theta}}(\boldsymbol{z}|\boldsymbol{x})}[p_{\boldsymbol{\theta}}(\boldsymbol{y}|\boldsymbol{x}, \boldsymbol{z})]$. Nevertheless, this joint distribution does not have a closed-form expression or efficient estimator, and thus, the result of this marginalization is not differentiable with respect to its parameters, which makes it unfeasible to optimize it as it could be done with a directed classifier. The intractability of $p_{\boldsymbol{\theta}}(\boldsymbol{y}|\boldsymbol{x})$ in a latent variable model is due to the intractability of $p_{\boldsymbol{\theta}}(\boldsymbol{z}|\boldsymbol{x})$, since $p_{\boldsymbol{\theta}}(\boldsymbol{y}|\boldsymbol{x}, \boldsymbol{z})$ is tractable. Therefore, since $p_{\boldsymbol{\theta}}(\boldsymbol{y}|\boldsymbol{x}, \boldsymbol{z})$ can be computed efficiently, $p_{\boldsymbol{\theta}}(\boldsymbol{y}|\boldsymbol{x})$ is tractable if and only if $p_{\boldsymbol{\theta}}(\boldsymbol{z}|\boldsymbol{x})$ is tractable. Nevertheless, we must find and approximation for the latter distribution.

## 3.2 Embeddings Extractor and Lower Bound

To approximate $p_{\boldsymbol{\theta}}(\boldsymbol{z}|\boldsymbol{x})$, we introduce a parametric model $q_{\boldsymbol{\phi}}(\boldsymbol{z}|\boldsymbol{x})$. This model is called embeddings extractor and $\boldsymbol{\phi}$ are the parameters of this model, also called variational parameters, which are optimized so that $q_{\boldsymbol{\phi}}(\boldsymbol{z}|\boldsymbol{x}) \approx p_{\boldsymbol{\theta}}(\boldsymbol{z}|\boldsymbol{x})$. This approach allows us to optimize $p_{\boldsymbol{\theta}}(\boldsymbol{y}|\boldsymbol{x})$ since, as in any other variational method, the objective to maximize the *evidence lower bound* (ELBO). In this case, as we detail in Appendix A.1, we formulate the ELBO as:

$$\mathcal{L}_{\boldsymbol{\theta}, \boldsymbol{\phi}}(\boldsymbol{x}, \boldsymbol{y}, \boldsymbol{z}) = \mathbb{E}_{q_{\boldsymbol{\phi}}(\boldsymbol{z}|\boldsymbol{x})}\left[\log \frac{p_{\boldsymbol{\theta}}(\boldsymbol{y}, \boldsymbol{z}|\boldsymbol{x})}{q_{\boldsymbol{\phi}}(\boldsymbol{z}|\boldsymbol{x})}\right] \tag{1}$$

We should note that we could introduce a variational model $q_{\boldsymbol{\phi}}(\boldsymbol{z}|\boldsymbol{x}, \boldsymbol{y})$, which for each input $\boldsymbol{x}$ would return as many embeddings as classes. However, in order to align with fully-observed classifiers, we design our embeddings extractor to obtain a single embedding per input.

Given the dataset $\mathcal{D}$ with distribution $q_\mathcal{D}(\boldsymbol{x}, \boldsymbol{y})$, the objective is to maximize the expectation of the ELBO under that distribution. Therefore, as demonstrated in Appendix A.2, this is equivalent to:

$$\mathbb{E}_{q_{\mathcal{D}(\boldsymbol{x}, \boldsymbol{y})}}\left[\mathcal{L}_{\boldsymbol{\theta}, \boldsymbol{\phi}}(\boldsymbol{x}, \boldsymbol{y}, \boldsymbol{z})\right] = -CE\left(q_\mathcal{D}(\boldsymbol{x}, \boldsymbol{y}), p_{\boldsymbol{\theta}}(\boldsymbol{y}|\boldsymbol{x})\right) - \mathbb{E}_{q_{\mathcal{D}(\boldsymbol{x}, \boldsymbol{y})}}\left[D_{KL}\left(q_{\boldsymbol{\phi}}(\boldsymbol{z}|\boldsymbol{x})||p_{\boldsymbol{\theta}}(\boldsymbol{z}|\boldsymbol{x})\right)\right] \tag{2}$$

where we assume that $p_{\boldsymbol{\theta}}(\boldsymbol{y}|\boldsymbol{x}) = \mathbb{E}_{q_{\boldsymbol{\phi}}(\boldsymbol{z}|\boldsymbol{x})}[p_{\boldsymbol{\theta}}(\boldsymbol{y}|\boldsymbol{x}, \boldsymbol{z})]$, since $q_{\boldsymbol{\phi}}(\boldsymbol{z}|\boldsymbol{x}) \approx p_{\boldsymbol{\theta}}(\boldsymbol{z}|\boldsymbol{x})$. The first term of this loss function is the cross-entropy used in the vanilla classifier and encourages the model to classify correctly. The second term regulates the way in which embeddings are distributed in space, so that it is close to the target distribution $p_{\boldsymbol{\theta}}(\boldsymbol{z}|\boldsymbol{x})$. The key here is that this distribution has many degrees of freedom, since it can depend on the input or some of its attributes and can either be prefixed before training or modified during training, since it depends on $\boldsymbol{\theta}$.

## 3.3 Stochastic Gradient Optimization of the ELBO

Calculating $\nabla_{\boldsymbol{\phi}} \mathbb{E}_{q_{\boldsymbol{\phi}}(\boldsymbol{z}|\boldsymbol{x})}[p_{\boldsymbol{\theta}}(\boldsymbol{y}|\boldsymbol{x}, \boldsymbol{z})]$ is problematic. To solve this, we can use the reparameterization trick Kingma & Welling (2013) to estimate this term and avoid calculating the exact gradient. To do so, we reparametrize the variable $\tilde{z} \sim q_{\boldsymbol{\phi}}(\boldsymbol{z}|\boldsymbol{x})$ using a differentiable transformation $\boldsymbol{g}_{\boldsymbol{\phi}}(\boldsymbol{\epsilon}, \boldsymbol{x})$

and an auxiliary random variable $\boldsymbol{\epsilon}$, such that $\tilde{z} = \boldsymbol{g_\phi}(\boldsymbol{\epsilon}, \boldsymbol{x}), \ \boldsymbol{\epsilon} \sim p(\boldsymbol{\epsilon})$. Then, we have:

$$p_{\boldsymbol{\theta}}(\boldsymbol{y}|\boldsymbol{x}) \approx \frac{1}{L} \sum_{l=1}^{L} p_{\boldsymbol{\theta}}\left(\boldsymbol{y}|\boldsymbol{x}, \boldsymbol{g_\phi}\left(\boldsymbol{\epsilon}^{(l)}, \boldsymbol{x}\right)\right), \ \boldsymbol{\epsilon}^{(l)} \in p(\boldsymbol{\epsilon}) \tag{3}$$

On the other hand, the second term inside the expectation in equation 2 can be calculated analytically in the cases we are going to work with, as explained in next section.

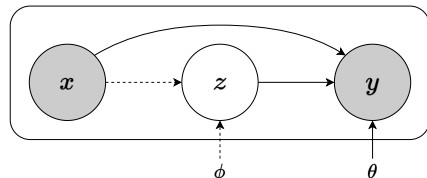

Figure 1: Directed graphical model. The dashed lines represent $q_\phi(\boldsymbol{z}|\boldsymbol{x})$ and the solid lines represent $p_{\boldsymbol{\theta}}(\boldsymbol{y}|\boldsymbol{x}, \boldsymbol{z})$. The parameters $\phi$ and $\boldsymbol{\theta}$ are learned jointly during training.

## 4   A VARIATIONAL CLASSIFIER

In this section we propose the use of neural networks to model the classifier as a latent variable model. To do so, we start by giving the neural network formulation of a classifier constructed as a fully-observed model, which we call vanilla classifier. In the vanilla classifier, we distinguish two parts. The first one is $\boldsymbol{h_\phi}$, which is the embeddings extractor and the second one is $\boldsymbol{f_\theta}$, which is the last layer and whose output contains the probabilities of each class. Given an input $\boldsymbol{x}$:

$$\boldsymbol{z} = \boldsymbol{h_\phi}(\boldsymbol{x}) \in \mathbb{R}^k \tag{4}$$

$$\boldsymbol{y} = \boldsymbol{f_\theta}(\boldsymbol{z}) \in \mathbb{R}^C \tag{5}$$

where $\boldsymbol{z}$ is the embedding of the input $\boldsymbol{x}$, $k$ is the embeddings dimension and $\boldsymbol{y}_j$ is the probability that $\boldsymbol{x}$, belongs to class $j \in \{1, 2, \ldots, C\}$. On the other hand, we can calculate the output of the Variational Classifier (VC) as:

$$\boldsymbol{z}^{(l)} = \boldsymbol{g_\phi}\left(\boldsymbol{x}, \boldsymbol{\epsilon}^{(l)}\right) \in \mathbb{R}^k, \ \boldsymbol{\epsilon}^{(l)} \in p(\boldsymbol{\epsilon}) \tag{6}$$

$$\boldsymbol{y} = \frac{1}{L \cdot N} \sum_{l=1}^{L} \sum_{n=1}^{N} \boldsymbol{f_\theta}\left(\boldsymbol{z}^{(l)}\right) \in \mathbb{R}^C \tag{7}$$

by employing neural networks, we have implemented the model described in equation 3, resulting in the last two expressions presented above, equation 4 and equation 5. Based on the previous derivation, it becomes possible to construct a diverse range of latent variable classifiers using neural networks. However, in our proposal, we introduce two specific conditions on the general model:

- First, we impose that $p_{\boldsymbol{\theta}}(\boldsymbol{z}|\boldsymbol{x}) = p_{\boldsymbol{\theta}}(\boldsymbol{z}|\boldsymbol{y}) = \mathcal{N}(\boldsymbol{\mu_\theta}(\boldsymbol{y}), \text{diag}(\boldsymbol{\sigma_\theta^2}))$. That is, we impose that the model $p_{\boldsymbol{\theta}}(\boldsymbol{z}|\boldsymbol{x})$ is a centered isotropic multivariate Gaussian whose mean depends on the class label of the input $\boldsymbol{x}$. We should note that we could make $\boldsymbol{z}$ depend on any attribute or feature of $\boldsymbol{x}$, but, since the goal is to classify its class, we make it depend only on it. In this way, once the model is trained, the embeddings of same-class entries obtained from $q_\phi$ will be close together in the space, which we argue will result in better performance. On the other hand, for simplicity, we assume the variance of the distribution to be independent of the class, although it could potentially vary depending on the classes as well.

- The second condition is that $q_\phi(\boldsymbol{z}|\boldsymbol{x}) = \mathcal{N}(\boldsymbol{\mu_\phi}(\boldsymbol{z}|\boldsymbol{x}), \text{diag}(\boldsymbol{\sigma_\phi}(\boldsymbol{z}|\boldsymbol{x})^2))$. In this case, we impose that the model $q_\phi(\boldsymbol{z}|\boldsymbol{x})$ is also a centered isotropic multivariate Gaussian and that in this case depends on the input.

In this case we can then define $\boldsymbol{g_\phi}(\boldsymbol{x}, \boldsymbol{\epsilon}^{(l)}) = \boldsymbol{\mu_\phi}(\boldsymbol{z}|\boldsymbol{x}) + \boldsymbol{\sigma_\phi}(\boldsymbol{z}|\boldsymbol{x}) \odot \boldsymbol{\epsilon}^{(l)}$ and $p(\boldsymbol{\epsilon}) = \mathcal{N}(0, \boldsymbol{I})$. In the next two sections we provide two ways of modeling $\boldsymbol{\mu_\phi}(\boldsymbol{z}|\boldsymbol{x})$ and $\boldsymbol{\sigma_\phi}(\boldsymbol{z}|\boldsymbol{x})$ using neural networks. We also provide the algorithms to train each of them in Appendix B.

### 4.1 Learnable Variance Variational Classifier (LVVC)

To model what was previously described, we first present a formulation in which we add two affine transformations $\boldsymbol{h}_{\phi_\mu}$ and $\boldsymbol{h}_{\phi_\sigma}$ such that:

$$\boldsymbol{\mu}_\phi(\boldsymbol{z}|\boldsymbol{x}) = \boldsymbol{h}_{\phi_\mu}(\boldsymbol{h}_{\phi'}(\boldsymbol{x})) \in \mathbb{R}^k \tag{8}$$

$$\boldsymbol{\sigma}_\phi(\boldsymbol{z}|\boldsymbol{x}) = \boldsymbol{h}_{\phi_\sigma}(\boldsymbol{h}_{\phi'}(\boldsymbol{x})) \in \mathbb{R}^k \tag{9}$$

Where $\boldsymbol{h}_{\phi'}$ is the architecture used in the vanilla classifier as embeddings extractor ($h_\phi$ in equation 4). We must note that this formulation adds a number of parameters that, for common values of $k$, is quite small with respect to the total number of parameters of the whole system. Specifically, the total number of trainable parameters added is $2k(k+1)$.

In the previous section we explained how to estimate the first term of the loss function. In this case, we can estimate the second term as (see Appendix C.1):

$$\mathbb{E}_{q_{\mathcal{D}(\boldsymbol{x},\boldsymbol{y})}} \left[ D_{KL} \left( \mathcal{N}(\boldsymbol{\mu}_\phi(\boldsymbol{z}|\boldsymbol{x}), \boldsymbol{\sigma}_\phi^2(\boldsymbol{z}|\boldsymbol{x}) \cdot I) || \mathcal{N}(\boldsymbol{\mu}_\theta(y), \boldsymbol{\sigma}_\theta^2 \cdot I) \right) \right]$$
$$\approx \frac{1}{2|\mathcal{D}|} \sum_{(\boldsymbol{x},\boldsymbol{y})\in\mathcal{D}} \sum_{j=1}^k \log \frac{\boldsymbol{\sigma}_\theta^2}{\boldsymbol{\sigma}_\phi^2(\boldsymbol{z}|\boldsymbol{x})_j} - 1 + \frac{(\boldsymbol{\mu}_\phi(\boldsymbol{z}|\boldsymbol{x})_j - \boldsymbol{\mu}_\theta(y)_j)^2}{\boldsymbol{\sigma}_\theta^2} + \frac{\boldsymbol{\sigma}_\phi^2(\boldsymbol{z}|\boldsymbol{x})_j}{\boldsymbol{\sigma}_\theta^2} \tag{10}$$

### 4.2 Fixed Variance Variational Classifier (FVVC)

One of the problems with the above formulation is that we need to slightly modify the original architecture of a vanilla classifier. However, by imposing some more conditions, we can keep the original architecture. This formulation is:

$$\boldsymbol{\mu}_\phi(\boldsymbol{z}|\boldsymbol{x}) = \boldsymbol{h}_\phi(\boldsymbol{x}) \in \mathbb{R}^k \tag{11}$$

$$\boldsymbol{\sigma}_\phi(\boldsymbol{z}|\boldsymbol{x}) = \boldsymbol{\sigma}_\phi = \boldsymbol{\sigma}_\theta \in \mathbb{R}^k \tag{12}$$

In this case, we assume that the variance of $q_\phi(\boldsymbol{z}|\boldsymbol{x})$ is constant and equal to the variance of the target distribution $p_\theta(\boldsymbol{z}|\boldsymbol{y})$ and that the embeddings extractor should estimate only the mean. Finally, it is trivial to see from equation 10, that the second term of the loss function can be approximated as:

$$\mathbb{E}_{q_{\mathcal{D}(\boldsymbol{x},\boldsymbol{y})}} \left[ D_{KL} \left( \mathcal{N}(\boldsymbol{\mu}_\phi(\boldsymbol{z}|\boldsymbol{x}), \boldsymbol{\sigma}_\phi^2(\boldsymbol{z}|\boldsymbol{x}) \cdot I) || \mathcal{N}(\boldsymbol{\mu}_\theta(\boldsymbol{y}), \boldsymbol{\sigma}_\theta^2 \cdot I) \right) \right]$$
$$\approx \frac{1}{2|\mathcal{D}|\boldsymbol{\sigma}_\theta^2} \sum_{(\boldsymbol{x},\boldsymbol{y})\in\mathcal{D}} \sum_{j=1}^k (\boldsymbol{\mu}_\phi(\boldsymbol{z}|\boldsymbol{x})_j - \boldsymbol{\mu}_\theta(\boldsymbol{y})_j)^2 \tag{13}$$

We should note that the fact that the model is not modified means that the only thing that really changes is the meaning of output of the penultimate layer, which, while in the vanilla classifier is an embedding, in this case it is the mean of the distribution followed by these embeddings. In addition, another advantage is that conditions can be imposed on the embeddings space, as we explain below.

### 4.3 Conditions on $\boldsymbol{\mu}_\theta$ and $\boldsymbol{\sigma}_\theta$

First, for the variance, we choose for simplicity that $\boldsymbol{\sigma}_\theta = \boldsymbol{1}$, i.e. the target distribution to be spherical with unit variance.

Regarding $\boldsymbol{\mu}_\theta$, the first option is not to impose any condition on it (uncond. in Table 1). However, we demonstrate in Appendix C.2 that in order to maximize $D_{KL} \left( \mathcal{N}(\boldsymbol{\mu}_j, \mathrm{diag}(\sigma^2)) || \mathcal{N}(\boldsymbol{\mu}_k, \mathrm{diag}(\sigma^2)) \right)$, we must minimize $\langle \boldsymbol{\mu}_j, \boldsymbol{\mu}_k \rangle$. Therefore, since we want our embeddings to be discriminative between classes, we must minimize the inner product between the means of the distributions of different classes. Since a priori we want to design a general method where the embeddings of all classes are equally separated from each other, we propose for simplicity the means of the distributions to be an orthogonal (or quasiorthogonal Kainen & Kůrková (1993; 2020)) set of vectors distinguishing between the next two cases:

$\boldsymbol{k} \geq \boldsymbol{C}$: Here we define a set of orthogonal vectors $\{\boldsymbol{u}_j\}_{j=1}^C$ and subsequently define $\boldsymbol{\mu}_\theta(\boldsymbol{y}_j) = \mu_\theta \cdot \boldsymbol{u}_j$, $j = 1, \ldots, C$, where $\mu_\theta \in \mathbb{R}$ is a learnable parameter.

$\boldsymbol{k} < \boldsymbol{C}$: In this case it is not possible to obtain a system of $C$ orthogonal vectors. However, we propose to use the Johnson-Lindenstrauss lemma Johnson (1984); Larsen & Nelson (2017) to obtain a set of $C$ vectors of length $k$ with a high distance between them. This lemma is defined as follows:

**Lemma 1** *For any $0 < \epsilon < 1$ and any integer $n$, let $k$ be a positive integer such that:*

$$k \geq 4(\epsilon^2/2 - \epsilon^3/3)^{-1} \ln n \tag{14}$$

*Then, for any set $V$ of $n$ points in $\mathbb{R}^d$, there is a map $f : \mathbb{R}^d \to \mathbb{R}^k$ such that for all $u, v \in V$:*

$$(1 - \epsilon)||u - v||^2 \leq ||f(u) - f(v)||^2 \leq (1 + \epsilon)||u - v||^2 \tag{15}$$

*where $f$ is called Johnson-Lindenstrauss Transform (JLT).*

This lemma states that a set of points in a high-dimensional space can be embedded in a lower dimensional space by a transformation such that the Euclidean distance between the points after applying this transformation are nearly preserved. Another interesting property of this lemma is that after applying a JLT $f$ to a set of points, the dot product is also nearly preserved. Corollary 1.1 is proved in Appendix D and formalizes this property.

**Corollary 1.1** *Let $\epsilon$, $V$, and $f$ be as defined in Lemma 1, and furthermore let $f$ be linear. Then for every $u$, $v \in V$, if $-v \in V$:*

$$\langle u, v \rangle - \epsilon ||u|| ||v|| \leq \langle f(u), f(v) \rangle \leq \langle u, v \rangle + \epsilon ||u|| ||v|| \tag{16}$$

Therefore, to obtain $\{\boldsymbol{\mu_\theta}(\boldsymbol{y}_j)\}_{j=1}^C$, we first define a set of $C$ orthonormal vectors of length $C$ and apply on all of them a linear JLT $f : \mathbb{R}^C \to \mathbb{R}^k$ to obtain a set of vectors $\{\boldsymbol{u}_j\}_{j=1}^C$. Subsequently, we define the set of means $\{\boldsymbol{\mu_\theta}(\boldsymbol{y}_j)\}_{j=1}^C$ in such a way that $\boldsymbol{\mu_\theta}(\boldsymbol{y}_j) = \mu_\theta \cdot \boldsymbol{u}_j$, where is $\mu_\theta \in \mathbb{R}$ is a learnable parameter. To obtain $f$ we use the algorithm proposed in Ailon & Chazelle (2009).

## 5 EXPERIMENTS

As we explain in the third paragraph of the 1 section, the system described above has three main advantages over other systems. In the following we describe the sets of experiments developed to demonstrate these three advantages.

### 5.1 CLASSIFICATION PERFORMANCE

To carry out this experiment we use residual networks He et al. (2016) as encoders trained on three well-known image datasets: CIFAR-10, CIFAR-100 and ImageNet and compared them with other methods.

**CIFAR-10/CIFAR-100** Krizhevsky (2009) are two datasets composed of 60000 color images of $32 \times 32$ having 10/100 classes with the same number of elements each. To perform the comparisons we train with different methods from the literature a ResNet-20 and ResNet-110 with CIFAR-10, and a ResNet-56 and ResNet-110 with CIFAR-100 (ResNets serve as $\boldsymbol{h_\phi}$ in equation 4). To study the performance of our VC, in addition to a vanilla classifier, we compare it with other well-known approaches in the literature. Among them, there are methods that include techniques that try to increase Euclidean distance Zhang et al. (2020) or angle Liu et al. (2018) between embeddings of different classes, focus on discriminating hard examples Lin et al. (2017); Ryou et al. (2019) or force intra-class orthogonality at the minibatch level Ranasinghe et al. (2021). The formulas we follow to train all methods are described in Appendices E.2 and E.3. We run all the experiments five times and present the mean values in Table 1. To train the VC, we use $L = 10$, since we have observed that for these datasets, increasing the number of samples offers no benefit.

**ImageNet** Krizhevsky et al. (2017) is a large-scale dataset containing about 1.2 million train images and 50,000 validation images belonging to 1,000 classes. We use a ResNet-34 as embeddings extractor¡ and in this case we compare our VCs only with a vanilla classifier that uses the cross-entropy as loss function, since this architecture is more expensive to train. Here we use $L = 100$ during training of VC, as it gives better results than $L = 10$, which may be due to the higher dimension of the latent space. We use the formula explained in Appendix E.4 to train all methods.

Table 1: Results of different loss functions used for classification with different configurations of the VC for CIFAR-10, CIFAR-100 and ImageNet. The evolution of the accuracy for some methods and datasets of interest are given in Appendix F. RNx is used to abbreviate ResNet-x.

| Loss | CIFAR-10 | | CIFAR-100 | | | | ImageNet | |
| --- | --- | --- | --- | --- | --- | --- | --- | --- |
| | RN20 | RN110 | RN56 | | RN110 | | RN34 | |
| | Top-1 | Top-1 | Top-1 | Top-5 | Top-1 | Top-5 | Top-1 | Top-5 |
| CE | 91.68 | 93.50 | 69.87 | 91.25 | 71.95 | 91.80 | 72.97 | 91.19 |
| Sphere Face | 91.68 | 93.61 | 69.07 | 89.74 | 69.85 | 91.02 | - | - |
| RBF | 91.28 | 93.40 | 70.13 | 91.05 | 71.83 | 91.63 | - | - |
| Focal Loss | 91.44 | 93.69 | 70.35 | 91.54 | 72.01 | 91.89 | - | - |
| Anchor Loss | 90.86 | 71.47 | 70.29 | 91.07 | 71.94 | 91.23 | - | - |
| OPL | 91.88 | 93.68 | 70.31 | **91.61** | 71.95 | 92.00 | - | - |
| LVVC ($\mu_\theta$ uncond.) | 91.73 | 93.63 | 69.98 | 91.41 | 71.68 | 91.90 | 73.38 | 91.22 |
| LVVC ($\mu_\theta$ orth.) | 91.81 | 93.71 | 69.86 | 91.09 | 72.02 | 91.87 | 73.47 | 91.44 |
| FVVC ($\mu_\theta$ uncond.) | 91.65 | 93.59 | **70.49** | 91.41 | 72.16 | 91.87 | 73.92 | **91.80** |
| FVVC ($\mu_\theta$ orth.) | **91.86** | **93.74** | 70.38 | 91.38 | **72.31** | **92.06** | **73.93** | 91.73 |

Table 1 shows that in all the scenarios there is a VC that improves with a certain margin over the rest of the systems analyzed with respect to the Top-1 Accuracy. However, analyzing the Top-5 Accuracy, we can observe that the difference with other systems is smaller and even surpassed in one case. This may be due to the fact that we have chosen unimodal target distributions, so that all embeddings of the same class tend to go to a single mode. However, when we use loss functions that do not impose restrictions on the embeddings, the distribution of the embeddings of the same class tends to be multimodal. Therefore, in the latter case, embeddings of two similar inputs but belonging to different classes are more likely to be in closer modes than in the VC with unimodal target distributions. This opens the door to the convenience of exploring different target distributions depending on the objective of the problem. On the other hand, we observe that, in addition, the FVVC performs better overall than the LVVC in terms of accuracy, which is advantageous as this design does not introduce additional parameters compared to the Vanilla Classifier. In addition, forcing the means of the target distributions of different classes to be orthogonal to each other raises in most cases the accuracy of the target distributions. It seems, therefore that imposing this condition enhances class separability, as theoretically explained.

## 5.2 CALIBRATED UNCERTAINTY

Another potential advantage of knowing the distribution of the embeddings is that more than one embedding can be sampled from that distribution and a prediction can be obtained from each embedding. This allows averaging from the predictions and obtaining a single better calibrated prediction. To verify this, we compare a VC (FVVC $\mu_\theta$ orth.) with a Vanilla Classifier and the following methods, which are references in the literature for obtaining calibrated predictions: **Ensemble** Lakshminarayanan et al. (2017) (Prediction from combined output of $M$ independently trained models with random initialization), **MC Dropout** Gal & Ghahramani (2016) (Dropout on the entire network during test with rate $p$), **LL-MC Dropout** Riquelme et al. (2018) (MC Dropout only on the activations before last layer) and **Temperature Scaling** Guo et al. (2017) (Post-hoc calibration by temperature using a validation dataset). The computational and storage cost of the latter two is comparable to that of the VC, while that of the former two is higher. This information is detailed in Appendix G.1. Specifically, we perform two comparisons for scenarios in which good calibration is particularly important and which are described below and based on Ovadia et al. (2019).

**Corrupted Data** In scenarios where the test data are corrupted or shifted with respect to the training data, the accuracy is expected to decrease. Ideally, however, the uncertainty of the predictions in this scenario should be higher. In order to study this, we train a ResNet-56 with CIFAR-100 and evaluate its accuracy and calibration for the original test data and modified versions of these with different types and levels of common corruptions in vision data. Specifically, we find 19 types of corruptions and 5 levels for each of them Hendrycks & Dietterich (2019). The objective of this analysis is to observe how the performance in terms of accuracy and Expected Calibration Error (ECE) DeGroot & Fienberg (1983); Naeini et al. (2015) evolves as the level of corruption increases. For the

ensemble method we have trained $N = 10$ models. For both VC and MC Dropout methods we have averaged over $K = 128$ predictions during test stage. In addition, we set the dropout rate to $p = 0.5$ in MC Dropout methods. Figure 2 shows the results of this comparison. The first observation is that the VC is better calibrated in general terms than methods of similar cost. However, the two more expensive methods have a lower level of ECE, which was to be expected. On the other hand, except for Ensemble, none of the other methods stand out significantly in terms of accuracy. We can conclude then that in this scenario VC is better calibrated than other methods with similar cost, but lags behind those that require more resources. In Appendix G.2 we perform an identical analysis to the present one for CIFAR-10 and a ResNet-20.

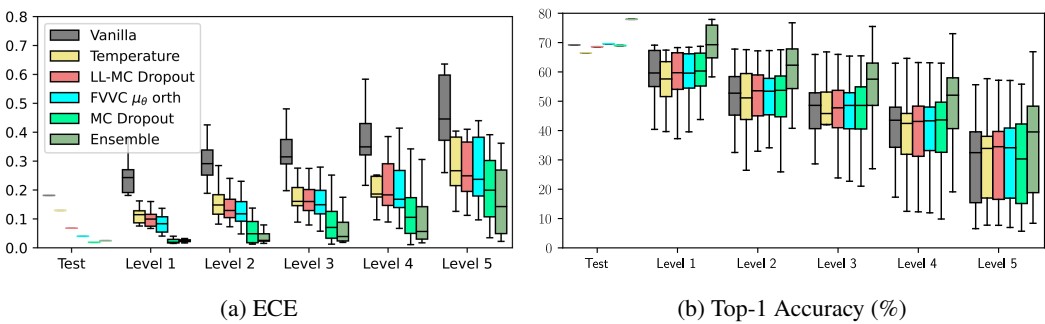

| (a) ECE | (b) Top-1 Accuracy (%) |

Figure 2: ECE and Top-1 Accuracy in the methods compared and for the different levels of corruption for CIFAR-100. Each box plot shows the quartiles across all corruption types for each level.

**Out of Distribution Detection** A well-designed classifier should be able to identify when the input data does not belong to any of the classes is has been trained on. This is called Out of Distribution (OOD) Detection and the more calibrated the predictions are, the better the performance in this task should be. Equivalently, this means that when the inputs to the model are not from any of the classes for which it has been trained, the predictions should have high uncertainty. To study the performance of the different alternatives in OOD, we train ResNet-20 with CIFAR-10 using the previously explained methods. Subsequently, in the test phase, we pass as input elements of the well-known SVHN dataset Netzer et al. (2011). As evaluation metrics we use confidence and entropy and show the results in Figure 3. In this case, it is of interest that the area under the curves in figures 3a 3b is as small as possible. That is, the ideal situation would be that all predictions would be uniformly distributed and therefore their entropy is as high as possible (close to $\log(10) \approx 2.3026$ in this case) for as many examples as possible. We see that all the compared methods are far from this behavior, but again the two more expensive ones perform better. For this purpose, Temperature Scaling outperforms our VC, but this still outperforms MC Dropout in the last layer and Vanilla Classifier. Similar results for a ResNet-56 trained with CIFAR-100 are presented in Appendix G.3.

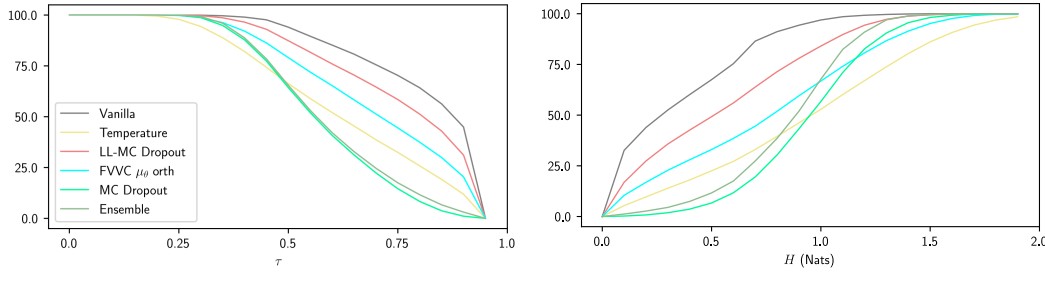

| (a) % of predictions with confidence greater than $\tau$ | (b) % of predictions with entropy lower than $H$ |

Figure 3: Confidence and entropy for predictions of an OOD dataset (SVHN) in a model trained with CIFAR-10.

### 5.3 BETTER SPACE EMBEDDINGS UTILIZATION

A phenomenon that has been described in Liang et al. (2022) and is not desirable is the so-called cone effect. This refers to the fact that when a model is trained following the usual procedures, the effective embedding space is restricted to a narrow cone and that embeddings are close together regardless of whether they correspond to similar input data or not. The fact that in the VC the target distributions of the embeddings depend on the class greatly reduces this phenomenon. To illustrate this, we train a ResNet-20 with MNIST LeCun et al. (1998) following the formula in E.1. In Figure 4 we show the average angle between embeddings of all the classes. We indeed see that, with all VCs this effect is reduced. Furthermore, in the FVVC, the embeddings of different classes end up being almost orthogonal even if this condition is not imposed. On the other hand, the LVVC achieves the orthogonality objective better than the FVVC when this condition is imposed. Moreover, when such condition is not imposed, the some angles between the embeddings of different classes are larger than in the unconditioned FVVC. In general, it seems that the FVVC has a poorer representation than the LVVC of the embedding space, which makes sense, since for the latter the variance of $q_\phi(z|x)$ is learnable.

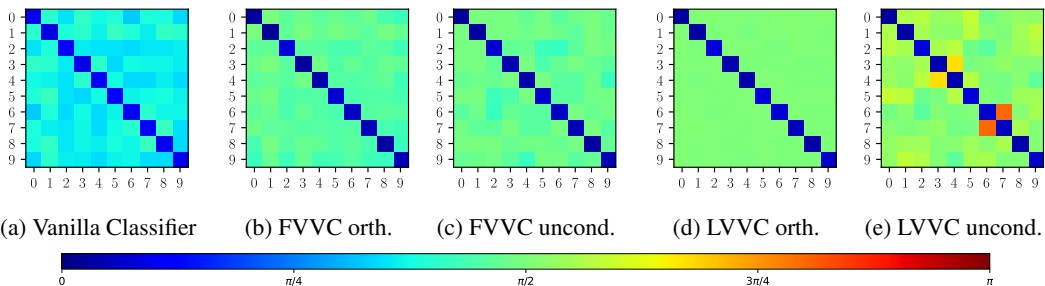

(a) Vanilla Classifier     (b) FVVC orth.     (c) FVVC uncond.     (d) LVVC orth.     (e) LVVC uncond.

Figure 4: Average angle between embeddings of each class for MNIST dataset.

## 6 CONCLUSION

In this paper we have presented a way to model a classifier as a latent variable model. This allows modeling the distribution of the latent variable, which corresponds to what is usually called embedding. It is therefore possible to sample over this distribution and obtain as many predictions as desired. The main advantage of this over fully-observed classifiers is that it allows the uncertainty of the predictions to be estimated.

In addition, to optimize this system we derive a lower bound which we maximize by using a stochastic approximation. The maximization of this lower bound achieves simultaneously (i) minimizing the difference between the labels and the predictions of the model and (ii) regularizing the distribution of embeddings in the space according to their class and input. In this second point lies the other great advantage of this classifier, which is that this derivation offers the possibility of imposing certain conditions on the embedding space.

In addition, we have proposed two implementations of this system using neural networks. The first one slightly modifies the architecture of a conventional classifier and the second one does not modify it at all. In these proposals we have shown how, in a simple way, it is possible to impose conditions on the embeddings space. In particular, we have imposed that embeddings of different classes be orthogonal or quasiorthogonal to each other, which improves the accuracy of the classifier in most of the experiments performed.

Finally, we have empirically demonstrated that our system is more accurate than other top-performing alternatives in the literature. Furthermore, we have proven that its predictions are better calibrated than those from other methods in the literature with comparable cost even when the inputs are corrupted or belong to another domain to which the classifier has been trained. Finally, we have shown that the space utilization of embeddings in our approach is better than in a conventional classifier.

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
