# OpenReview forum: "Variational Bayes Classifier"
_ICLR.cc/2024/Conference — ICLR 2024 Conference Withdrawn Submission_

### Official Review · Reviewer_pf4E · 2023-10-25

**Soundness:** 3 good
**Presentation:** 3 good
**Contribution:** 2 fair
**Rating:** 5
**Confidence:** 4

**Summary:**

The paper presents an alternative method for learning classifiers instead of the standard cross-entropy loss. The authors propose a latent variable model to learn a distribution over the embeddings of the penultimate layer. Similar to variational-auto encoders (VAEs) the authors suggest an amortized inference with NNs to output the mean and possibly the standard deviation of each dimension of the latent variable. Here, instead of using a standard Gaussian prior, the authors suggest a more informative prior which may depend on the class label. The authors compare their method to baselines for learning embedding and Bayesian baselines on several tasks.

**Strengths:**

The paper presents the following merits in my opinion:
* The idea of using a latent variable model for classification tasks is novel as far as I am ware.
* I liked the idea of using an informative prior that contributes to inter-class separation and intra-class clustering.
* The method has a similar computational demand as standard NN based classifiers.
* The paper is easy to follow and understand.
* Code was provided and the method seems to be reproducible. Exact experimental details were given.

**Weaknesses:**

Several issues that I found with this submission:
* I strongly disagree with the claim presented in the paper that this direction can be or in some sense is a replacement for Bayesian methods.
  * First, and it may be a manner of personal opinion, one cannot obtain a reliable uncertainty estimation in the predictions without accounting for the uncertainty in (or at least in some of) the parameters. The empirical evaluation of this method does not show otherwise. Both MC-dropout which is considered a relatively weak Bayesian baseline and deep ensemble which is quite old by now significantly outperform LVVC/FVVC in that aspect. I acknowledge that the suggested approach improves over vanilla training, but it does not suffice in my opinion to showcase the claim in the paper.
  * Second, empirical evidence shows that similar approaches to the one suggested in this paper do not model well the uncertainty. Specifically, Bayesian VAEs are used in [1] for OOD detection. And, in [3] it is shown on standard classification tasks that latent variable models combined with deep kernels [2] do not capture well the uncertainty of the model.
* The accuracy gains in Table 1 seem to be marginal. Perhaps, the results are statistically significant, but it is hard to tell without knowing the variance of them. I suggest adding information about the variance in order to get a better sense. Also, there seems to be a mistake in the first column (RN20-top1), OPL is slightly better than LVVC and FVVC.
* There are some inaccuracies in the writing:
  * In the related work section you refer to general latent variable models as mean-field. If I understand the intention correctly, this is not true.
  * From the paper: "The intractability of $p_θ(y|x)$ in a latent variable model is due to the intractability of $p_θ(z|x)$, since $p_θ(y|x, z)$ is tractable." To me, it sounds like the reason that $p_θ(y|x)$ is intractable is because $p_θ(z|x)$ is intractable, but this doesn't have to be the case. Both the posterior and the likelihood can be tractable but $p_θ(y|x)$ may still be intractable (for instance, a Bernoulli likelihood with a Gaussian posterior).
* Minor:
  * To make the submission complete I think the algorithm for obtaining the Johnson-Lindenstrauss Transform should be added to the appendix.
  * The inline citation format is odd - it is missing parentheses.
  * Why does FVVC perform better than LVVC? I think a proper explanation or analysis is missing to clarify that.

[1] Daxberger, E., & Hernández-Lobato, J. M. (2019). Bayesian variational autoencoders for unsupervised out-of-distribution detection. arXiv preprint arXiv:1912.05651.
[2] Liu, H., Ong, Y. S., Jiang, X., & Wang, X. (2021). Deep latent-variable kernel learning. IEEE Transactions on Cybernetics, 52(10), 10276-10289.
[3] Achituve, I., Chechik, G., & Fetaya, E. (2023). Guided Deep Kernel Learning. arXiv preprint arXiv:2302.09574.

**Questions:**

.

---

### Official Review · Reviewer_QwG7 · 2023-10-27

**Soundness:** 2 fair
**Presentation:** 2 fair
**Contribution:** 2 fair
**Rating:** 5
**Confidence:** 3

**Summary:**

The authors propose a variational bayes classifier. The stated goal is to enable estimation of model uncertainty while avoiding the computational cost associated with methods such as Bayesian Neural Nets. Posing classification through a latent variable model also allows for conditions to be imposed on the learned embeddings, such as interclass separation. The authors provide an ELBO to optimize the parameters of the model using SGD. They also propose to versions of their method.

**Strengths:**

- The paper is original in the sense that it constructs a fully supervised variational model as opposed to the close relatives in the unsupervised and semi-supervised VAE cases. Here the emphasis is on learning classes and uncertainty rather than reconstruction of the input data.
- The paper addresses a significant problem. Being able to provide reliable uncertainty estimates around classification is a very important task as it enables trust in ML models.

**Weaknesses:**

The paper can be improved as well as the contributions can be clarified further on multiple aspects including originality, quality, clarity as well as significance. Specific questions are provided in the Questions sections but in general:
- the paper seems to make broader claims than it demonstrates in the experiments.
- the quality of experiments section/figures can be significantly improved, both from visual perspective as well as intepretation.
- the relationship of some of the concepts to existing variational bayes concepts should be clarified, to show originality.
- the significance of using this particular method over existing methods isn't quite clear.

**Questions:**

1: Please discuss how Equation 1, 2 is different from / similar to existing variational bayes objectives, particularly the objective used for semi-supervised VAE [1, 2]

[1] Semi-Supervised Learning with Deep Generative Models, Kingma et al
[2] SHOT-VAE: Semi-supervised Deep Generative Models with Label-aware ELBO Approximations Feng et al

2: In the abstract it's claimed the the paper demonstrates "empirically the theoretical advantages of our proposal using different architectures..." . This statement might set reader up to expect theorems that predict these advantages, that are then demonstrated empirically. This doesn't seem to be the case.

3: What's the point of FVVC? given the modification suggested in LVVC doesn't seem to be particularly difficult to implement in conventional NNs, why give up the ability to learn uncertainty when that's a significant part of your work?

4: 4th line after in Sec 4.3, please specify what 'j' and 'k' stand for.

5: This is particularly important. The results in Table 1 hardly seem to be significant improvements over other methods, in fact in some cases they are bolded even though they are not the highest (e.g. CIFAR-10 RN20 Top-1, OPL is 91.88, FVVC is bolded at 91.86, in most other cases improvements are of less than 0.5%). Can we really draw broad stroke conclusions such as those made in the paragraph under Table 1 with such insignificant differences? A paper that is focused on uncertainty of predicted labels can definitely appreciate how such minor differences in accuracy may not be worth drawing conclusions from. Very interested in author's perspective on this. Same goes for results presented in Figure 2.

6: It seems like the only way to actually get uncertainty around predicted labels is to sample from embeddings and then get a prediction from all those samples. Doesn't that result in the same problem that was earlier mentioned for MC Dropout by Gal et al? "however, the prediction time is considerably longer, since it is necessary to pass the inputs to the model several times to obtain well-calibrated predictions". Please clarify.

7: Figure 3 is very hard to read, the color choices make it impossible to differentiate different curves, please consider making figures in the whole paper more visually differentiable.

8: How is the matrix of average angles between embeddings calculated for vanilla classifiers?

Some Suggestions:
Consider using the abbreviation "VBC" instead of VC to differentiate from "vanilla classifiers".
Typo in 4th line in sec 3.2 ("the objective IS to...")
Typo in 4th to last line in section 5.4 ("the some angles"... the is not needed)
I don't see a reference to Figure 1 in the main text... Also consider moving the Figure further up and add a description in the text.

---

### Official Review · Reviewer_KP2i · 2023-10-28

**Soundness:** 3 good
**Presentation:** 2 fair
**Contribution:** 2 fair
**Rating:** 5
**Confidence:** 4

**Summary:**

This paper introduced an approach for representing a classifier as a latent variable model using neural networks. Furthermore, they establish a lower bound for the optimization of this framework and provide empirical evidence of the efficacy of their model.

**Strengths:**

Clarity

- The paper is well written, making it comprehensible to readers.
- The suggested model and loss function are straightforward and effectively explained for easy comprehension.

Originality and Significance

- This paper introduces a novel method for depicting a classifier as a latent variable model employing neural networks.
- They provide empirical evidence that the proposed loss model results in enhanced generalization performance when compared to baseline methods.

**Weaknesses:**

Novelty
- Utilizing a hidden representation as the mean of a latent model with a fixed variance or transforming it with a linear model to determine the mean and variance of the latent model may be considered somewhat lacking in novelty, especially for the top conferences like ICLR.

Experiments
- I suggest that the authors include more recent models, such as efficient ensembling methods [1,2,3], as baseline comparisons and related works.
- It would be beneficial if the authors could present results showcasing the superiority of the embeddings trained by their proposed model compared to other models. They could potentially validate this empirically through transfer learning scenarios.

References

[1] Havasi, M., Jenatton, R., Fort, S., Liu, J. Z., Snoek, J., Lakshminarayanan, B., Dai, A. M., and Tran, D. Training independent subnetworks for robust prediction. In International Conference on Learning Representations (ICLR), 2021.

[2] Antoran, J., Allingham, J. U., and Hernandez-Lobato, J. M. Depth uncertainty in neural networks. In Advances in Neural Information Processing Systems 33 (NeurIPS2020), 2020.

[3] EungGu Yun, Hyungi Lee, Giung Nam, and Juho Lee. Traversing between modes in function space for fast ensembling. In International Conference on Machine Learning, 2023.

**Questions:**

See the weakness section.

---

### Official Review · Reviewer_fLRd · 2023-11-01

**Soundness:** 3 good
**Presentation:** 3 good
**Contribution:** 2 fair
**Rating:** 3
**Confidence:** 3

**Summary:**

This work presents the probabilistic classifier by assuming the feature of penultimate layer (last-layer) as the stochastic variable, and then modeling its mean and standard deviation of stochastic variable via neural network. Authors demonstrate the effectiveness of the proposed model by using the resnet with CIFAR 10 and CIFAR 100 datasets. Additionally, authors demonstrate that the latent hidden features of the proposed model using MNIST set can contain the meaningful information (orthogonality).

**Strengths:**

### Prior distribution of modeling via the label
The proposed model seems to be a variant of VAE for classification problem. The noticeable difference seems how to model the prior distribution of the latent variable $z$; authors models $p(z|x)$ by the NN mapping via the label $y$. This part seems a novel part of the proposed model.

**Weaknesses:**

### Weak Motivation

> Introducing a stochastic latent variable to handle the uncertainty issue for deterministic NN seems to be a weak motivation, especially when taking into account the current advancements in deep learning. To persuade the significance of  of this work, this work needs to explain why incorporating a stochastic latent variable is not only necessary but also offers more advantages compared to recent Bayesian deep learning approaches and other methods designed to handle uncertainty issue for deterministic NN.


###  Weak Contribution
> The proposed model seems to be a variant of VAE, tailored for classification.I think that there is similar approach presented in [1, 2]. If the valid reasons for the proposed approach, such as classifying the dataset with spurious correlation issue [2], are not properly explained, the contribution of the model itself seems weak.


[1] Uncertainty Estimation with a VAE-Classifier Hybrid Model - ICASSP 22

[2] Chroma-VAE: Mitigating Shortcut Learning with Generative Classifiers - NeurIPS 22

**Questions:**

* I am skeptical about whether the NN mapping of label $y$ yields a reasonable prior distribution $p(z|x)$ because the weight parameters of this NN mapping would require training to yield the meaningful prior distribution $p(z|x)$. Have the authors trained the weight parameters of the NN mapping used for the prior distribution $p(z|x)$ ? If so, what is a training objective for the weight parameters of NN mapping ? could you provide details on how to train these weight parameters?


* For $p_{\theta}(z | x)$, affine transformation $h_{\phi}$ in Eq. (8) and (9), the proposed model seems to use additional parameters compared to baseline models. How large are the NNs that are used to model $p_{\theta}(z | x)$ and two affine transformation $h_{\phi}$ in Eq. (8) and (9) ? As considering that additional parameters are used, the performance improvement in Table 1 and Figure 2 looks incremental.